# The Morphopathogenetic Aspects of Intraabdominal Adhesions in Children under One Year of Age

**DOI:** 10.3390/medicina55090556

**Published:** 2019-08-31

**Authors:** Anna Junga, Māra Pilmane, Zane Ābola, Olafs Volrāts

**Affiliations:** 1Institute for Anatomy and Anthropology, Rīga Stradiņš University, Riga LV-1010, Latvia; 2Department of Pediatric Surgery, Rīga Stradiņš University, Riga LV-1010, Latvia

**Keywords:** adhesion formation, growth factors, cytokines, enzymes, antimicrobial peptides, immunohistochemistry

## Abstract

*Background and Objectives:* The morphopathogenesis of adhesions is a complex process, characterized by the accumulation of an extracellular matrix, inflammation and hypoxia. The regulatory role between morphopathogenic factors in adhesions has not yet been defined. The aim was to investigate the appearance of transforming growth factor beta (TGFβ), basic fibroblast growth factor (FGF-2), fibroblast growth factor receptor 1 (FGFR1), protein gene product 9.5 (PGP 9.5), chromogranin A (CgA), interleukin-1 alpha (IL-1α), interleukin-4 (IL-4), interleukin-6 (IL-6), interleukin-7 (IL-7), interleukin-8 (IL-8), interleukin-10 (IL-10), tumor necrosis factor alpha (TNFα), matrix metaloproteinase-2 (MMP-2) and matrix metaloproteinase-2 tissue inhibitor (TIMP-2) in intraabdominal adhesions. *Materials and Methods:* The study material was obtained from 49 patients under one year of age with total or partial bowel obstruction. All factors were detected using immunohistochemistry methods and their relative distribution was evaluated by means of the semiquantitative counting method. *Results:* Intraabdominal adhesions are characterized by increased TGFβ, FGFR1 and decreased FGF-2, PGP 9.5, IL-1, IL-4, IL-8, TIMP-2 findings. The most significant changes observed were the remodulation of the extracellular matrix, promotion of neoangiogenesis and the maintenance of a prolonged inflammation. *Conclusions:* The increase in TGFβ, as well as the disbalance between MMP-2 and TIMP-2 proves an increased fibrosis in intraabdominal adhesions. Less detected FGF-2 and more prominent FGR1 findings points out a compensatory receptor stimulation in response to the lacking same factor. The decrease in PGP 9.5 indicate hypoxic injury and proves the stimulation of neoangiogenesis. An unpronounced IL-1 and marked IL-10 finding indicate the local tissue protection reaction, the decrease in IL-4 could be the direct cause of giant cells, but the decrease of IL-8 could confirm a delayed chemotaxis of inflammatory cells.

## 1. Introduction

Intraabdominal adhesions are defined as fibrous connections that develop between any surfaces in the peritoneal cavity [1]. Typically, adhesions occur after intraabdominal surgery and peritonitis [2,3], but rarely can also be congenital in their origin [4,5]. The pathogenesis of intraabdominal adhesions is a complex process, characterized by the accumulation of an extracellular matrix, tissue hypoxia and inflammation [6,7]. Complex interaction of several morphopathogenetic factors, which determine and modulate tissue growth, may play a role in the development of adhesions. Nowadays, we can study the morphopathogenetic processes at the cellular level. Therefore, the aim of this study was to evaluate and describe the morphopathogenetic factors that are involved in the formation and development of adhesions.

Transforming growth factor beta (TGFβ) is the most studied growth factor in the context of adhesions so far, as it regulates the proliferation and differentiation of cells, embryonic development, wound healing and angiogenesis. TGFβ regulates fibrotic processes by suppressing fibrinolysis [8]. Others growth factors have been also investigated—basic fibroblast growth factor (FGF-2) exerts pro-angiogenic actions via activation of fibroblast growth factor receptor 1 (FGFR1) [9], while increased levels of FGFR1 causes defects in wound healing [10].

Nerve fibers are identified in peritoneal adhesions, but general neuroendocrine system markers (protein gene product 9.5 (PGP 9.5), chromogranin A (CgA)) have not been identified yet.

The interaction between inflammatory cytokines in case of intraabdominal adhesions is unclear, because of lacking complex data. Mainly there are studies about postoperative adhesions in adults or experimental animals, but there are few studies about adhesions in children. Interleukin-1 (IL-1) is a major pro-inflammatory cytokine and pathogenic mediator of autoimmune, autoinflammatory, infectious and degenerative diseases [11]. The reduction of IL-1 and tumor necrosis factor alpha (TNFα) suppresses inflammation and the formation of adhesions [12,13,14,15]. Interleukin-4 (IL-4) is the key cytokine in the T helper type 2 differentiation, activating fibroblast proliferation and collagen production, thereby promoting development of fibrosis [16]. Interleukin-6 (IL-6) regulates acute phase responses [17] and correlates with the accumulation of collagen [18]. Interleukin-8 (IL-8) is one of the most potent neutrophil chemo-attractants in acute inflammation [19] and it poses appropriate conditions for fibrosis [20]. Interleukin-7 (IL-7) promotes synthesis of IL-4 and interleukin-10 (IL-10) [21]. IL-10 is a major anti-inflammatory [22] and important adhesion limiting factor [23].

Matrix metalloproteinase-2 (MMP-2) is an enzyme that breaks down different components of the extracellular matrix, for example type IV collagen, fibronectin and elastin [6,24]. Tissue inhibitor of metalloproteinase-2 (TIMP-2) regulates activity of MMP-2 and it has multiple effects on cell growth, differentiation and apoptosis [25]. Studies about the significance of matrix metalloproteinases and its tissue inhibitors are of high interest, because the balance between these enzymes regulates formation and degradation of the extracellular matrix [26].

The aim of this research was therefore to investigate the appearance and relative distribution of TGFβ, FGF-2, FGFR1, PGP 9.5, CgA, IL-1, IL-4, IL-6, IL-7, IL-8, IL-10, TNFα, MMP-2 and TIMP-2 in specimens of patients with intraabdominal adhesions.

## 2. Materials and Methods

The research work was done in accordance with the Helsinki declaration. The study was approved by the Ethical Committee at Rīga Stradiņš University, permit issued on 10 May 2007.

The study tissue material was obtained from 49 patients who underwent abdominal surgery due to complete or partial bowel obstruction. Six out of the 49 patients underwent repeated surgery, which is why together 57 tissue specimens were collected. After initial microscopic evaluation, 50 specimens were rated as appropriate for morphological analysis. A total of 21 out of 50 specimens were evaluated as congenital adhesions (embryonic peritoneal adhesions, Ladd band), but 29 as acquired adhesions related to gastrointestinal perforation, diffuse peritonitis and repeated surgeries. Most frequently adhesions were localized between the jejunal small intestinal loops and the proximal parts of ileum (21 cases) or at the duodenal region (ten cases). In four cases specimens were obtained from the distal parts of the ileum. In 13 cases adhesions were forming a Ladd band, but in another two cases the anterior abdominal wall was involved.The control group was obtained from eight patients with surgical repair of an inguinal hernia. Detailed patient data are summarised in table 1 and 2 of our previous paper [27].

The tissue material was fixed in a Stefanini solution [28]. The fixed tissue material was dehydrated in an alcohol solution of increasing concentration (from 70° to 96°) and degreased in a xylol solution. Subsequently tissues were embedded in paraffin. Three to four micrometers thin tissue cuts were prepared by means of a microtome (Leica RM2245, Leica Biosystems Richmond Inc., Buffalo Grove, IL, USA) and put on slides (HistoBond^®^+, Paul Marienfeld GmbH & Co. KG, Lauda-Königshofen, Germany). To create a general overview of the morphological picture, the slides were processed for the routine histological staining method. The stained preparations were examined using a light microscope (Leica DM500RB, Leica Biosystems Richmond Inc., Buffalo Grove, IL, USA).

### 2.1. Immunohistochemical Analysis

Tissue immunohistochemical staining for biomarkers identification was done by the biotin-streptavidin method [29], using the following antibodies:TGFβ (code orb7087, rabbit, working dilution 1:100, Biorbyt Ltd., Cambridge, UK),FGF-2 (code ab16828, rabbit, working dilution 1:200, Abcam, Cambridge, UK),FGFR1 (code ab10646, rabbit, working dilution 1:100, Abcam, Cambridge, UK),PGP 9.5 (code 439273A, rabbit, working dilution 1:100, ZYMED Laboratories, Invitrogen Corporation, Carlsbad, CA, USA),CgA (code 910216A, rabbit, working dilution 1:100, Invitrogen Corporation, Carlsbad, CA, USA),IL-1α (code sc-9983, mouse, working dilution 1:50, Santa Cruz Biotechnology, Inc., Dallas, TX, USA),IL-4 (code orb10908, rabbit, working dilution 1:100, Biorbyt Ltd., Cambridge, UK),IL-6 (code LS-B1582, mouse, working dilution 1:50, LifeSpan BioSciences, Inc., Seattle, WA, USA),IL-7 (code orb48420, rabbit, working dilution 1:100, Biorbyt Ltd., Cambridge, UK),IL-8 (code orb39299, rabbit, working dilution 1:100, Biorbyt Ltd., Cambridge, UK),IL-10 (code ab34843, rabbit, working dilution 1:400, Abcam, Cambridge, UK),TNFα (code sc-52250, mouse, working dilution 1:100, Santa Cruz Biotechnology, Inc., Dallas, TX, USA),MMP-2 (code orb11061, rabbit, working dilution 1:100, Biorbyt Ltd., Cambridge, UK), andTIMP-2 (code sc-21735, mouse, working dilution 1:50, Santa Cruz Biotechnology, Inc., Dallas, TX, USA) antibodies.

All antibodies used in research were diluted with Antibody Diluent (code-938B-05, Cell Marque^TM^, Rocklin, CA, USA).

Subsequently adhesion tissue cuts were deparaffinized and washed in alcohol and water, then rinsed with TRIS buffer (code-2017X12508, Diapath S.p.A., Martinengo, Italy) twice for five minutes and put in a boiling EDTA buffer (code-2017X02239, Diapath S.p.A., Martinengo, Italy) in a microwave for up to 20 min. When the samples had cooled down, they were washed twice for five minutes in a TRIS wash buffer. Further, blocking for ten minutes in 3% peroxide solution was performed, then washed twice for five minutes in TRIS wash buffer. All tissue samples were incubated with primary antibodies for one hour.

A HiDef Detection^TM^ HRP polymer system was used for the mice or rabbit origin antibodies. After the primary antibody incubation and rinsing with the TRIS wash buffer solution three times, the HiDef Detection^TM^ (code 954D-31, Cell Marque^TM^, Rocklin, CA, USA) reaction amplificator was applied for ten minutes at room temperature. Another washing for five minutes in TRIS wash buffer was performed. Further, incubation for ten minutes at room temperature with the HiDef Detection^TM^ HRP Polymer Detector (code-954D-32, Cell Marque^TM^, Rocklin, CA, USA) was performed. Again, rinsing with TRIS buffer solution was performed three times for five minutes. After this processing, tissue coating with a liquid DAB Substrate Kit chromogenic system (code 957D-60, Cell Marque^TM^, Rocklin, CA, USA) up to ten minutes at room temperature was performed, resulting in positive structures staining brown.

Regardless of the staining system, after incubation with chromogenic substrate system samples were rinsed with running water and counterstaining with hematoxylin (code-05-M06002, Mayer’s Hematoxylin, Bio Optica Milano S.p.A., Milano, Italy) was performed. Finally, all samples were dehydrated with increasing concentration alcohols (from 70° to 96°) and clarified with carboxylic acid and xylol. Then a slide adhesive for coverslip attachment was used.

For each preparation series, positive controls of the tissues indicated by the manufacturer, which always have a positive reaction, were prepared. As negative control, the parallel cuts of the preparation, where the primary antibody was substituted by the antibody diluent Antibody Diluent, were used.

Slides were examined under a light microscope (Leica DM500RB, Leica Biosystems Richmond Inc., Buffalo Grove, IL, USA). Findings were fixed with a Leica DC 300F camera and analyzed with the image-processing software Image Pro Plus.

The semiquantitative counting method was used for the registration of the relative amount of immunopositive structures [30]. The designations were as follows:0—no positive structures in the visual field;0/+—occasionally positive structures in the visual field;+—few positive structures in the visual field;+/++—few to moderate positive structures in the visual field;++—moderate positive structures in the visual field;++/+++—moderate to numerous positive structures in the visual field;+++/++++—numerous to abundant positive structures in the visual field;++++—abundant positive structures in the visual field.

The amount of structures was analyzed in five fields of view of a randomly selected section. The average amount of structures was chosen for further analysis.

### 2.2. Data Analysis

To characterize the research group descriptive statistic methods were used. For the description of each marker, the median and interquartile range was used. For the comparison of groups, Mann–Whitney U-tests were used [31]. To evaluate the cross-compliance of two variables, Spearman’s rank correlation coefficient (rs) was calculated [32]. The acquired results were interpreted: rs ≤ 0.35—weak correlation, 0.35 < rs < 0.7—moderate correlation, and rs ≥ 0.7—strong correlation. Two-tailed *p* values of <0.05 were considered as statistically significant. Statistical analysis was conducted using the Statistical Package for the Social Sciences (SPSS) program version 23.0 (IBM Corporations, Armonk, NY, USA).

## 3. Results

Abnormal tissue changes were observed in all adhesion tissue specimens. Modifications in mesotheliocytes (Figure 1a) and fibroblasts (Figure 1b) were characteristics of adhesion tissue. Chaotically placed dense connective tissue fiber bundles (Figure 1c) and sometimes also large collagen fiber accumulations without fibroblasts (Figure 1d) were observed. In almost all specimens, adhesion tissues were infiltrated by neutrophils and macrophages. In one part of the patient’s specimen, inflammatory cells were located diffusely, in the other part—perivascularly (Figure 1e), often around sclerotized arterioles. In the case of a diffuse marked inflammation we also found epithelioid cells (Figure 1f). In the overview specimen, we frequently observed hyperemic blood vessels and neoangiogenesis (Figure 1g).

A moderate (++) number of TGFβ-positive structures in the adhesions group was found. Mostly TGFβ-positive fibroblasts and macrophages (Figure 2a) were detected. Sometimes, also epithelioid cells and endotheliocytes were seen. Compared to the control group (Figure 2b), statistical tests confirmed that significantly more TGFβ-positive structures (U = 50.5, *p* < 0.001) were found in the adhesions group.

The FGF-2 findings in the adhesion tissues were variable—from occasional (0/+) to abundant (++++) positive structures. A positive reaction was seen in fibroblasts and macrophages (Figure 2c). In the adhesion specimen, positive structures were found in significantly lower number for FGF-2 than in control tissues (Figure 2d; U = 83.0, *p* = 0.007).

A moderate (++) number of FGFR1-positive structures in the adhesion group was detected. Most often FGFR1-positive fibroblasts and macrophages (Figure 2e) were observed, as well as positive endotheliocytes in some of the specimens. Compared to the control group (Figure 2f), FGFR1-positive structures were seen in a significantly higher number in the adhesion group (U = 81.00, *p* = 0.006). A statistically significant, moderately tight positive correlation was found between FGF-2 and FGFR1 (rs = 0.490, *p* < 0.001).

In the adhesions, PGP 9.5-positive nerve fibers and shape modified fibroblasts (Figure 3a) were most often observed in few to moderate (+/++) or moderate (++) appearance. Compared to the control group (Figure 3b), a statistically significant lower amount of PGP 9.5-positive structures in adhesions was proven (U = 58.5, *p* = 0.001). Statistically significant, moderately tight positive correlations were detected between PGP 9.5 and TIMP-2 (rs = 0.524, *p* < 0.001) and PGP 9.5 and MMP-2 (rs = 0.515, *p* < 0.001). A statistically significant negative correlation was found between PGP 9.5- and TGFβ-positive structures (rs = −0.386, *p* = 0.007).

CgA was visible in endotheliocytes, fibroblasts and sometimes in macrophages (Figure 3c). Mostly separate (0/+) or few (+) CgA-positive structures were detected. The amount of the observed structures was not significantly different in comparison to the control group tissues (Figure 3d; U = 170.5, *p* = 0.491).

A few to moderate (+/++) number of IL-1α-positive macrophages (Figure 4a) and fibroblasts (also structurally changed fibroblasts) were seen in the adhesions. Significantly less IL-1α-positive structures were found in the adhesion group compared to the control group (Figure 4b); U = 98.5, *p* = 0.015).

IL-4-positive structures (fibroblasts and macrophages (Figure 4c) in the adhesion group were mostly observed in few (+) or few to moderate (+/++) appearance. IL-4 findings in the adhesions were significantly lower than in the control group tissues (Figure 4d; U = 60.5, *p* = 0.002). Statistically significant positive correlations were observed between IL-4 and MMP-2 (rs = 0.490, *p* < 0.001), as well as between IL-4 and TIMP-2 (rs = 0.320, *p* = 0.026).

In the adhesions mostly a moderate (++) amount of IL-6 positive inflammatory cells (neutrophils, macrophages) and fibroblasts, as well as structurally changed fibroblasts were seen. In some specimens IL-6-positive epithelioid cells (Figure 4e) were found. The number of positive structures in the adhesions and control group (Figure 4f) did not differ (U = 146.5, *p* = 0.243).

Generally, numerous (+++), moderate to numerous (++/+++) or moderate (++) number of IL-7-positive fibroblasts, macrophages, endotheliocytes (Figure 4g) and epithelioid cells were detected in the adhesions. There was no statistically significant difference between the groups (Figure 4h; U = 144.5, *p* = 0.190).

The IL-8 finding in the adhesion group tissues was variable. Mostly few (+) fibroblasts and macrophages contained IL-8, but some specimens showed few to moderate (+/++), moderate (++) (Figure 4i) or moderate to numerous (++/+++) positive structures. The amount of IL-8 found in adhesions was significantly lower (U = 40.0, *p* < 0.001) in comparison to the control group (Figure 4j).

In the adhesion group IL-10 was observed in macrophages, epithelioid cells, neutrophils and fibroblasts (Figure 4k). Generally, IL-10-positive structures were seen in moderate to numerous (++/+++) amounts, which was not statistically different from the control group (Figure 4l; U = 184.0, *p* = 0.769).

In the adhesions group tissues, a moderate (++) number of TNFα-positive fibroblasts and macrophages (Figure 4m) were detected. In eight specimens, positive epithelioid cells were seen. There was no statistically significant difference between the groups (Figure 4n; U = 124.0, *p* = 0.082).

Using the Spearman’s correlation test, statistically significant, moderately tight positive correlations were found between inflammation regulatory factors: IL-7 and IL-4 (rs = 0.491, *p* < 0.001), IL-1 and IL-7 (rs = 0.471, *p* = 0.001), IL-7 and IL-8 (rs = 0.440, *p* = 0.001), IL-1 and IL-10 (rs = 0.438; *p* = 0.002), TNFα and IL-6 (rs = 0.436, *p* = 0.002), IL-10 and IL-7 (rs = 0.433, *p* = 0.002), IL-1 and IL-4 (rs = 0.365, *p* = 0.010) and IL-1 and IL-6 (rs = 0.325, *p* = 0.023).

MMP-2-positive inflammatory cells, mainly macrophages and neutrophils, epithelioid cells (Figure 5a), fibroblasts and endotheliocytes were mostly detected in few to moderate (+/++) number. There was no statistically significant difference between the groups (Figure 5b; U = 174.0, *p* = 0.654). A statistically significant negative correlation was detected between MMP-2 and TGFβ (rs = -0.333, *p* = 0.021).

The TIMP-2 finding in the adhesions group tissues was variable. Few (+) to moderate to numerous (++/+++) fibroblasts and inflammatory cells (Figure 5c) contained TIMP-2. Significantly less TIMP-2-positive structures were seen in the adhesion group compared to the control group (Figure 5d). The most distinct difference was detected between TIMP-2-positive fibroblasts (U = 108.00, *p* = 0.022) and endotheliocytes (U = 34.0, *p* < 0.001).

All semiquantitative results are summarized in Table 1.

## 4. Discussion

The morphopathogenesis of intraabdominal adhesions is not fully understood, but it is known that fibrin matrix formation and disbalance of fibrinolysis, resulting from mesothelial injury, appears to play role in the adhesion formation. Activation of pro- and anti-inflammatory factors may interact extensively with the fibrinolytic pathway. Unfortunately, there are only few studies about the development of adhesions and their mechanisms in children, especially in newborns.

TGFβ is so far the most investigated growth factor in the pathogenesis of adhesions. We detected a moderate number of TGFβ-positive structures in most adhesion specimen. Evaluating the TGFβ findings in adhesions tissues, we noticed significantly more TGFβ-positive structures compared to the control group, indicating a possible role in the morphopathogenesis of intraabdominal adhesions. This proves of an ongoing stimulation of adhesion forming structures, which is supported by our finding of large accumulations of collagen bundles. It is depicted in literature, that TGFβ regulates the formation of the extracellular matrix [33]. The function of TGFβ is characterized by an increased synthesis of extracellular matrix components [34] and increased expression of TGFβ is related to fibrosis in various tissues, thereof in peritoneal adhesions [35]. TGFβ inhibits the activity of matrix metalloproteinases, therefore, not only does it stimulate the accumulation of extracellular matrix components, but also delays its degeneration [36]. We detected a negative correlation between TGFβ and MMP-2 findings, which is consistent with previous reports in the literature.

Interestingly, in one-third of the adhesion specimens, there were very few, occasional or no FGF-2-positive structures observed. It is thought that the statistically significant decrease in FGF-2-positive cells, is responsible for fibrosis to occur. FGF-2 reduces scarring and supports wound healing, through inhibiting the differentiation of myofibroblasts [37]. This factor also inhibits myofibroblast precursors and connective tissue fibroblasts [38]. FGF-2 has a high affinity for binding FGFR1 [39]. We observed a statistically significant positive correlation between FGF-2 and FGFR1, however there were significantly higher numbers of FGFR1-positive structures seen in the adhesion group. The decreased in FGF-2 and more pronounced FGFR1 finding proves a compensatory receptor stimulation in response to the lack of the same factor, in case of adhesions.

PGP 9.5 is widely used as a marker for innervation and neuropeptide containing structures. The innervation in intraabdominal adhesions has not been investigated much, so far, and the role of PGP 9.5 in intraabdominal tissues is unclear. Most often few to moderate or moderate PGP 9.5-positive nerve fibers and shape-modified fibroblasts were observed and a significantly lower amount of PGP 9.5-positive structures in adhesion tissues was proven. PGP 9.5 is necessary for axon stability and its loss causes axon degeneration and neuronal cell death [40,41]. A decrease in PGP 9.5 is a characteristic finding in ischemic injury [41]. It cannot be excluded that the decrease in PGP 9.5 in intraabdominal adhesions is due to tissue hypoxia. In our study, we observed a statistically significant positive correlation between PGP 9.5 and the fibrosis promoting TGFβ, as well as MMP-2 and TIMP-2—factors responsible for connective tissue matrix remodulation. This is supported by the fact that PGP 9.5-positive fibroblast-like cells, promoting fibrosis, were found in hepatic fibrosis specimen [42].

Publications about the significance of CgA in fibrotic processes are sporadic and the appearance of CgA in adhesions is still unclear. In our study, we detected occasional CgA-positive structures, without significant differences in comparison to the control group. However, the Italian investigator Angelo Corti reported that CgA regulates the formation of stroma in tumor tissues, promoting a shift of fibroblast phenotype and production of extracellular matrix proteins [43].

Until now only some of the interleukins are described in case of adhesions, but a systematic analysis on pro- and anti-inflammatory cytokines and their appearance in intraabdominal adhesions are lacking. IL-1 is the main inflammatory cytokine [11]. It promotes adhesions, increasing fibrin accumulation and inhibiting fibrinolysis [44]. A few to moderate number of IL-1-positive cells were seen in adhesions and this finding was significantly lower compared to control group tissues. Therefore, we think of an ineffective systemic inflammation process. We observed that the IL-1 finding statistically significantly correlated with other interleukins (IL-4, IL-6, IL-7), proving of its role in the induction of pro-inflammatory cytokines in case of adhesions. The pathogenesis of intraabdominal adhesions could be related to local defense reactions. The overall moderate to numerous IL-10-positive cells and their correlations with other pro-inflammatory mediators makes one think of an ongoing local protection process. IL-10 inhibits inflammation, mainly through reducing the synthesis of pro-inflammatory mediators (IL-1, IL-6, IL-8, TNFα) [45].

IL-4 supports tissue remodeling processes, it is the main stimulator of an anti-inflammatory response in macrophages [46]. IL-4 stimulates the formation of macrophage aggregates or the development of giant multinuclear cells [47]. In our study, we observed giant multinuclear cells in almost 20% of cases. In our opinion, IL-4 may be the initiator of giant epithelioid cell formation. Overall, we detected significantly lower number of IL-4-positive structures in adhesion tissues. We found a weak positive correlation between IL-4 and TIMP-2, but a moderately tight positive correlation between IL-4 and MMP-2 in intraabdominal adhesions. We assume that the lack of IL-4 and a disbalance in MMP-2/TIMP-2 maintains the adhesion process.

IL-8 is one of the most important cytokines, promoting neutrophil leukocyte activation and their migration to the place of inflammation [48]. In the adhesion group, mostly few or few to moderate number of IL-8-positive structures were seen, which was significantly less than in the control group. Possibly, such a finding indicates a delay in neutrophil leukocyte chemotaxis in case of adhesions, which prolongs the course of inflammation. This is indirectly supported by the finding of inflammatory cell infiltration and the occurrence of epithelioid cells in adhesion tissues.

Overall, in our study we observed a moderate number of IL-6 and moderate to numerous numbers of IL-7-positive inflammatory cells (neutrophils, macrophages) and fibroblasts, but in both cases we did not detect a statistically significant difference between the groups. In our opinion IL-6 and IL-7 can maintain tissue inflammation, however these cytokines are non-specific in the pathogenesis of intraabdominal adhesions. An increase of IL-6 has been observed in the peritoneal fluid in patients after abdominal surgery [49]. IL-1 and TNFα are main activators of IL-6 expression [50]. This fact explains the complex coherencies we observed—a positive correlation between IL-6 and IL-1, as well as IL-6 and TNFα immunoreactive structures. IL-7 promotes the activation of fibroblasts and macrophages [51]. We observed positive correlation between IL-7 and IL-1, IL-4, IL-8 and IL-10. This finding is consistent with other author’s findings reporting the stimulation of IL-7 secretion through other cytokines, such as IL-10 and TNFα [52].

In our study, positive macrophages and fibroblasts for TNFα were seen in moderate numbers in the examined specimens. Despite the lack of a statistically significant difference between our study groups, predominantly moderate appearances of TNFα-positive cells in the adhesions makes it thinkable that this factor is involved in pathogenesis of intraabdominal adhesions. Hypoxia causes normal peritoneal fibroblasts to acquire an adhesion phenotype, which exhibited significantly higher levels of TNFα compared to normal peritoneal fibroblasts [53]. The role of TNFα in the pathogenesis of postoperative adhesion formation has been proven by histopathologic evaluation of rat model after treatment with TNFα inhibitors. Decreased levels of TNFα were associated with a reduction in adhesion formation [13].

The balance between MMPs and TIMPs expression regulates normal wound healing. Changes in expression levels can cause the formation of excess tissue, such as the development of adhesions [54]. We observed few to moderate number of MMP-2-positive structures in the adhesions, but we did not detect statistically significant differences to the control group. Interestingly, we observed a moderate number of TIMP-2-positive structures, however, in the adhesion group we observed significantly less TIMP-2-positive endotheliocytes and fibroblasts, pointing out the disbalance between MMP-2/TIMP-2. Our data are consistent with other author’s findings, regarding the fact, that the increased MMP-2 activity and decreased TIMP-2 expression significantly increases the relation between MMP-2/TIMP-2, promoting accumulation of collagen and fibrosis [55]. It cannot be excluded that human beta-defensin-2 (HBD-2) also promotes the disbalance between MMP-2/TIMP-2, increasing extracellular matrix synthesis. In our previous study we detected moderate to numerous numbers of HBD-2 positive structures in adhesions [27]. Statistically significant positive correlations were observed between HBD-2, MMP-2 and TIMP-2. The correlation between HBD-2 and MMP-2 was strong, but between HBD-2 and TIMP-2 moderately strong.

At the end of the discussion, it has to be stressed that the complex morphopathogenetic processes of intraabdominal adhesion development in infants have been analyzed for the first time in our study. Therefore, all together it can be concluded that intraabdominal adhesions in children are characterized by increased TGFβ, FGFR1 and decreased FGF-2, PGP 9.5, IL-1, IL-4, IL-8, TIMP-2 findings.

## 5. Conclusions

The increase in TGFβ-containing structures, as well as disbalance between MMP-2 and TIMP-2 in adhesion tissues gives evidence of the growth/regenerative potential of loose connective tissue and proves increased fibrosis in the pathogenesis of intraabdominal adhesions.Less detected FGF-2 and more prominent FGR1 findings point out a compensatory receptor stimulation in response to the lacking same factor in adhesion tissues.The decrease in PGP 9.5-positive structures indicate hypoxic injury and tissue ischemia and proves the stimulation of neoangiogenesis.An unpronounced IL-1 and marked IL-10 finding indicate the dominating local tissue protection reaction, the decrease in IL-4-positive structures could be the direct cause of giant cells, but the decrease of IL-8-positive structures could confirm a delayed chemotaxis of inflammatory cells and the prolongation of the inflammatory process.Similar findings of CgA in both groups show the unspecific role of these factors in morphopathogenesis of adhesions.

## Figures and Tables

**Figure 1 medicina-55-00556-f001:**
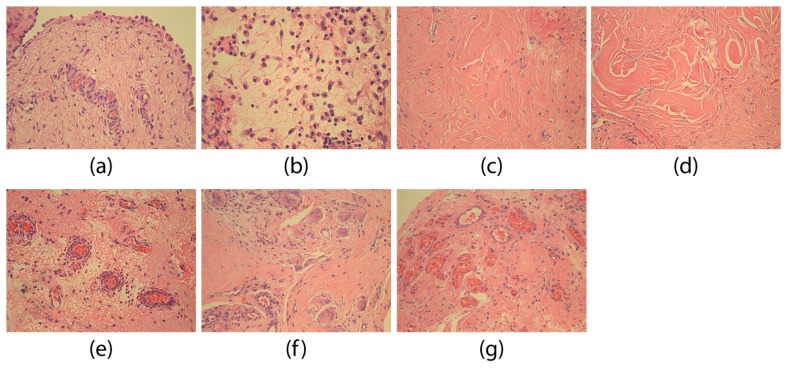
The morphological peculiarities of adhesions. (**a**) Section of adhesion tissues with aberrant round-shaped mesotheliocytes of a 151-day-old patient. H&E, 250×; (**b**) aberrant round-shaped fibroblasts in a 14-day-old patient. H&E, 400×; (**c**) chaotically located dense connective tissue fiber bundles in the adhesion specimen of a 100-day-old patient. H&E, 200×; (**d**) chaotically located dense connective tissue bundles and collagen fiber accumulations without fibroblasts in a 100-day-old patient. H&E, 250×; (**e**) perivascular inflammation in the adhesions of a 129-day-old patient. H&E, 250×; (**f**) epithelioid cells in a 56-day-old patient. H&E, 200×; and (**g**) neoangiogenesis in a 100-day-old patient. H&E, 200×.

**Figure 2 medicina-55-00556-f002:**
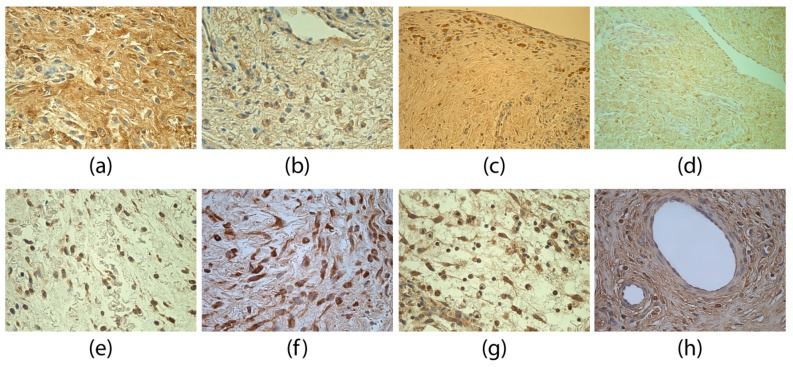
Immunoreactive structures for TGFβ, FGF-2 and FGFR1. (**a**) Numerous TGFβ-positive fibroblasts, macrophages and connective tissue fibers in adhesions of a 39-day-old patient. TGFβ IMH, 400×; (**b**) few to moderate TGFβ-positive fibroblasts and macrophages of a 76-day-old patient in the control group. TGFβ IMH, 400×; (**c**) moderate FGF-2-positive fibroblasts and macrophages in adhesions of a four-day-old patient. FGF-2 IMH, 400×; (**d**) numerous FGF-2-positive fibroblasts of a 145-day-old patient in the control group. FGF-2 IMH, 400×; (**e**) numerous FGFR1-positive fibroblasts and macrophages in adhesions of a two-day-old patient. FGFR1 IMH, 400×; and (**f**) few FGFR1-positive fibroblasts of a 56-day-old patient in the control group. FGFR1 IMH, 400×.

**Figure 3 medicina-55-00556-f003:**
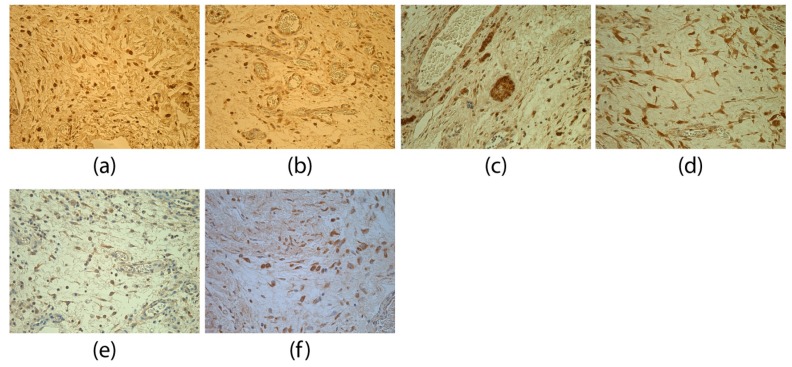
Immunoreactive structures for PGP 9.5 and CgA. (**a**) moderate PGP 9.5-positive nerve fibers and shape-modified fibroblasts in adhesions of a two-day-old patient. PGP 9.5 IMH, 250×; (**b**) moderate to numerous PGP 9.5-positive fibroblasts of a 76-day-old patient in the control group. PGP 9.5 IMH, 250×; (**c**) moderate CgA-positive fibroblasts and macrophages in adhesions of a 14-day-old patient. CgA IMH, 250x; and (**d**) few to moderate CgA-positive fibroblasts of a 76-day-old patient in the control group. CgA IMH, 250×.

**Figure 4 medicina-55-00556-f004:**
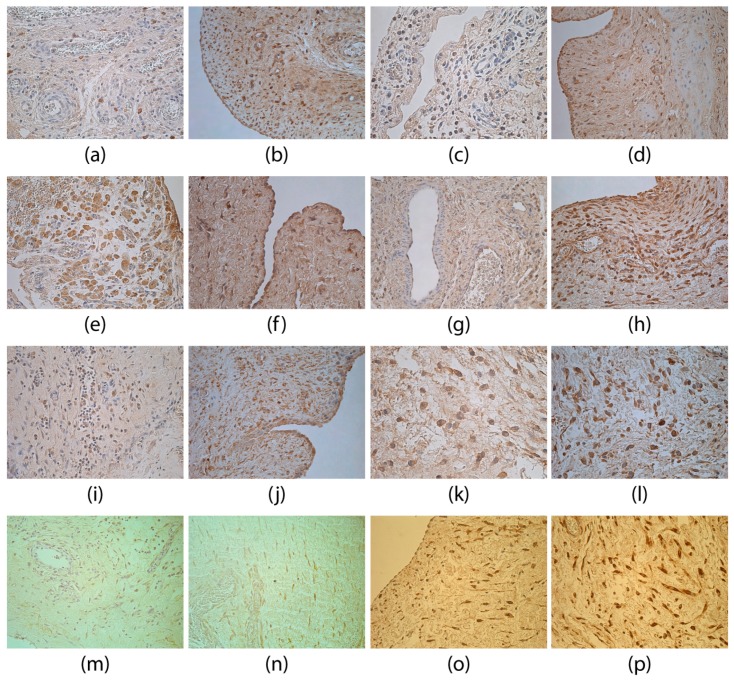
Immunoreactive structures for IL-1α, IL-4, IL-6, IL-7, IL-8, IL-10, TNFα and HBD-2. (**a**) Few IL-1α-positive macrophages in adhesions of a 14-day-old patient. IL-1α IMH, 250×; (**b**) moderate IL-1α-positive mesotheliocytes and fibroblasts of a 53-day-old patient in the control group. IL-1α IMH, 250×; (**c**) few to a moderate number of weakly IL-4-positive fibroblasts and macrophages in adhesions of a 30-day-old patient. IL-4 IMH, 400×; (**d**) numerous IL-4-positive mesotheliocytes and moderate fibroblasts of a 92-day-old patient in the control group. IL-4 IMH, 250×; (**e**) numerous IL-6-positive macrophages and epithelioid cells in adhesions of a two-day-old patient. IL-6 IMH, 250×; (**f**) moderate to numerous IL-6-positive mesotheliocytes of a 145-day-old patient in the control group. IL-6 IMH, 400×; (**g**) moderate IL-7-positive endotheliocytes, weakly positive fibroblasts and macrophages in adhesions of a 19-day-old patient. IL-7 IMH, 250×; (**h**) numerous IL-7-positive fibroblasts and macrophages of a 76-day-old patient in the control group. IL-7 IMH, 250×; (**i**) a moderate number of weakly IL-8-positive fibroblasts and macrophages in adhesions of a 56-day-old patient. IL-8 IMH, 250×; (**j**) moderate to numerous IL-8-positive mesotheliocytes and fibroblasts of a 145-day-old patient in the control group. IL-8 IMH, 250×; (**k**) moderate IL-10-positive fibroblasts and macrophages in adhesions of a patient less than a day old. IL-10 IMH, 400×; (**l**) moderate to numerous IL-10-positive macrophages and fibroblasts of a 46-day-old patient in the control group. IL-10 IMH, 400×; (**m**) moderate TNFα-positive fibroblasts and macrophages in adhesions of a 56-day-old patient. TNFα IMH, 250×; (**n**) moderate TNFα-positive fibroblasts and endotheliocytes of a 76-day-old patient in the control group. TNFα IMH, 250×.

**Figure 5 medicina-55-00556-f005:**
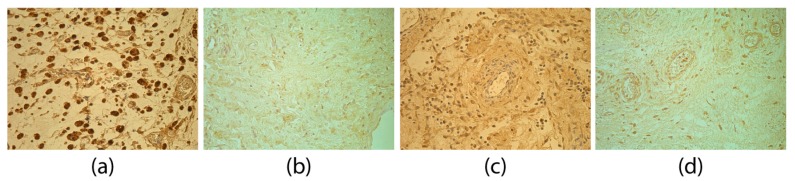
Immunoreactive structures for MMP-2 and TIMP-2. (**a**) Numerous MMP-2-positive macrophages and epithelioid cells in adhesions of a two-day-old patient. MMP-2 IMH, 250×; (**b**) moderate MMP-2-positive fibroblasts and macrophages of a 73-day-old patient in the control group. MMP-2 IMH, 250×; (**c**) moderate TIMP-2-positive fibroblasts, macrophages and endotheliocytes in adhesions of a 56-day-old patient. TIMP-2 IMH, 250×; and (**d**) moderate TIMP-2-positive fibroblasts and endotheliocytes of a 76-day-old patient in the control group. TIMP-2 IMH, 250×.

**Table 1 medicina-55-00556-t001:** Semi-quantitative evaluation of immunoreactive structures.

Factor	Group	Median	IQR	U	*p*
TGFβ	Adhesions	++	0.5	50.5	<0.001
	Control	+/++	0		
FGF-2	Adhesions	+	2	83.0	0.007
	Control	++/+++	0		
FGFR1	Adhesions	++	1.125	81.0	0.006
	Control	+	0.375		
PGP 9.5	Adhesions	+/++	0.5	58.5	0.001
	Control	++/+++	0.5		
CgA	Adhesions	+	0.5	170.5	0.491
	Control	+	0		
IL-1α	Adhesions	+/++	1.5	98.5	0.015
	Control	++	0		
IL-4	Adhesions	+/++	1	60.5	0.002
	Control	++	0.5		
IL-6	Adhesions	++	1	146.5	0.243
	Control	++/+++	0.5		
IL-7	Adhesions	++/+++	1	144.5	0.190
	Control	++/+++	0.5		
IL-8	Adhesions	+ to +/++	1	40.0	< 0.001
	Control	++/+++	0		
IL-10	Adhesions	++/+++	0.5	184.0	0.769
	Control	++/+++	0.5		
TNFα	Adhesions	++	0.5	124.0	0.082
	Control	++	0		
MMP-2	Adhesions	+/++	1	174.0	0.654
	Control	+/++	0.75		
TIMP-2	Adhesions	++	0.5	112.0	0.036
	Control	++	0		

Abbreviations: TGFβ—Transforming growth factor beta; FGF-2—basic fibroblast growth factor; FGFR1—fibroblast growth factor receptor 1; PGP 9.5—Protein gene product 9.5; CgA—Chromogranin A; IL-1α—Interleukin-1 alpha; IL-4—Interleukin-4; IL-6—Interleukin-6; IL-7—Interleukin-7; IL-8—Interleukin-8; IL-10—Interleukin-10; TNFα—Tumor necrosis factor alpha; MMP-2—Matrix metalloproteinase-2; TIMP-2—Tissue inhibitor of metalloproteinase-2; IQR—Interquartile range; U—Mann–Whitney U value; *p*—*p* value. Quantification of Structures: 0, no positive structures in the visual field; 0/+, occasionally positive structures in the visual field; +, few positive structures in the visual field; +/++ few to moderate positive structures in the visual field; ++, moderate positive structures in the visual field; ++/+++, moderate to numerous positive structures in the visual field.

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
