# Peer review of "The Morphopathogenetic Aspects of Intraabdominal Adhesions in Children under One Year of Age"

_medicina, 2019, doi:10.3390/medicina55090556_

Round 1
Reviewer 1 Report
The manuscript " The Morphopathogenetic Aspects of Intraabdominal Adhesions in Children under One Year of Age" shows new and interesting results, and it appears as a good work well structured. However major revision are needed.
· The authors have to improve the literature in the introduction and well better emphasize the aim of the work.
· At the line 84 is reported that “After initial microscopic evaluation 50 specimens were rated as appropriate for morphological analysis.” But not is reported with wich criteria.
· Being an immunohystological study the authors have to introduce the antibodies controls.
· Also images of antibodies expression in control tissues are needed.
· The authors can better explain the method used for the quantification (the quantification was performer only on one slide of one sample?The quantification was performer counting the number of positive cells?was made a discrimination between different cells like fibroblasto, macrophages, epithelioid cells and endotheliocytes).
· Moreover, since the specimens were obtained from different parts of the intestine (small intestine; distal ileum; duodenum; Ladd’s band; anterior abdominal wall) the expression of the different markers used in the work appears the same in all parts?
· Improve the description of the figures indicating the different cells expressing (fibroblasto, macrophages, epithelioid cells and endotheliocytes) each marker used in the study. Moreove, how did you have discriminate fibroblasto, macrophages, epithelioid cells and endotheliocytes?
· In the lines 188-194 is reported “Chaotically placed dense connective tissue fiber bundles and sometimes also large collagen fiber accumulations without fibroblasts were observed. In almost all specimen adhesion tissues were infiltrated by neutrophils and macrophages. In one part of the patient’s specimen inflammatory cells were located diffusely, in the other part – perivascularly, often around sclerotized arterioles. In the case of a diffuse marked inflammation we also found epithelioid cells. In the overview specimen, we frequently observed hyperaemic blood vessels and neoangiogenesis.”. Since this i san immunohystological study it could be good insert images showing all these alterations reported in the text.
Reviewer 2 Report
Authors in this manuscript identify pathological biomarkers in intraabdomianl adhesions in newborns. Using IHC approaches, they characterize markers of extracellular matrix, inflammation and hypoxia. They conclude that there was an increase TGF-β, FGFR1, VEGF and decrease of FGF-2, HGF, PGP 9.5, IL-1, IL-4, IL-8 and TIMP-2 by semi quantitative approach. However, there are few minor points to be addressed
Authors should investigate the role of activation evolutionarily conserved pathways in intraabdominal adhesions. For example markers of wnt and hippo pathways should be investigated Authors should also discuss the possible implications in clinical management/therapy derived from their findings
Round 2
Reviewer 1 Report
Dear authors,
the manuscript is now ready for publication.